# Perspectives of Licorice Production in Harsh Environments of the Aral Sea Regions

**DOI:** 10.3390/ijerph191811770

**Published:** 2022-09-18

**Authors:** Botir Khaitov, Aziz Karimov, Jamila Khaitbaeva, Obidjon Sindarov, Akmal Karimov, Yongqing Li

**Affiliations:** 1International Center for Biosaline Agriculture, Regional Office for Central Asia and South Caucasus, Tashkent 100084, Uzbekistan; 2Department of Soil Science and Farming, Tashkent Institute of Irrigation and Agricultural Mechanization Engineers, Tashkent 100000, Uzbekistan; 3Key Laboratory of South China Agricultural Plant Molecular Analysis and Genetic Improvement & Guangdong Provincial Key Laboratory of Applied Botany, South China Botanical Garden, Chinese Academy of Sciences, Guangzhou 510650, China

**Keywords:** Aral Sea region, harsh environment, licorice production, cultivation technology, sustainability

## Abstract

Along with pharmacological applications due to bioactive elements such as flavonoids and glycyrrhizin, licorice has positive influences on the rehabilitation, rejuvenation, and management of salt-affected degraded lands in arid regions. These features made this plant widely appreciated worldwide when climate change is showing detrimental impacts for crop production and food security. However, a growing demand followed by irrational harvesting of wild licorice plants has led to substantial dwindling of its natural habitat. There is an increasing need to protect the plant biodiversity since sustainability can be a problem with wild harvesting. Therefore, it is important to investigate cultivation technologies of licorice under harsh environments, while this plant can adapt to a wide range of climates. Thus, in this review, we studied, analyzed and summarized the literature on licorice cultivation methods counteracting the most common environmental stresses in the Aral Sea region. Particularly, the current knowledge was rationalized regarding on cultivation technologies for alleviating salt stress thereby improving crop production. We also highlighted that future research directions on licorice breeding and genomics that might facilitate to produce more resilient and sustainable licorice genotypes to renovate agricultural productivity under disastrous ecology and climate change of the arid regions. Whereas this area possesses all prerequisite conditions needed for successful cultivation of the alternative cash crop.

## 1. Introduction

Heavy exploitation of water from Amu Darya and Syr Darya rivers has eventually led to the shrinking of the Aral Sea. Therefore, this area is suffering catastrophic environmental degradation due to the intensification of soil salinity, a major threat to agricultural sustainability. Frequent droughts happening in the past years are also facilitating the development of soil salinity. Besides salinization, contamination by heavy metals and chemical compounds released by agriculture destroyed of the ecosystem in the surrounding region. These factors ultimately created a vicious cycle of environmental degradation. As a result, huge area of salinized lands was abandoned, destroying livelihoods that depended on them. This area comprises nearly 6 million hectares situated in Karakalpakstan, in the northwestern part of Uzbekistan, has been allocated to restore agricultural production. The severe impacts of land salinization to the ecosystem of the region were known for a at least half of century. Secondary salinization, soil erosion, and desertification are the main forms of land degradation in this area. The main drivers of land degradation are inappropriate water and soil management practices, as well as wind and water erosion, deforestation, and excessive grazing of livestock [1]. Water shortages and recurrent droughts exacerbate the problems of land degradation, resulting in decreased productivity and increased rural poverty [2]. It should be noted that the country has an agro-ecological and socio-economic system that is extremely dynamic and diversified, making it impossible to regulate the various forms of land degradation (Table 1).

The Aral Sea disaster obliged to shift the regional conventional farming into a climate-smart agricultural system. More recently, the growing of stress-resistant crops has become a top priority, maintaining sustainable production systems in this area. Since the climate is predicted to become more arid, advanced research technologies are being tested for trait discovery and large-scale field evaluation of stress-resistant crops. One of the current restoration initiatives in this region is the cultivation of licorice–a halophyte plant that grows in soil with a high salt concentration, so this practice can rehabilitate salinized soils and improve their fertility [2].

Licorice (*Glycorrhiza glabra*) belongs to the family Leguminosae and it is a native perennial plant in the Aral Sea region. Therefore, it grows naturally in wild habitats even under harsh conditions. The plant is quite unpretentious with a deep root system reaching up to 17 m deep, which plays an important role in reclamation and management of saline soils. Licorice is one of the few plants that has a wide range of uses in medicine, cosmetics, industry, and other industries. Licorice extracts have been used for medicinal purposes more than 2500 years [3]. The roots and stolons of licorice are collected for the major component, the sweet-flavored glycyrrhizin, an oleanane-type triterpene saponin. The flavonoids isolated from licorice are also used to prepare cosmetics [4]. In addition, a variety of components including sugars, flavonoids, sterols, amino acids, resins, starch, essential oil and saponins are derived from licorice roots [5].

Natural reserves of licorice are located in limited areas in Uzbekistan. Due to the over-exploitation, its distribution area has greatly decreased in recent years. Nowadays, natural areas of licorice can be found in the deltas of the Amudarya river (on the territory of the Republic of Karakalpakstan and Khorezm province) and Syrdarya river and on the banks of small rivers in Fergana valley, Surkhandarya and Syrdarya provinces of Uzbekistan [6]. Notably, the dwindling of the natural habitats of wild licorice in Uzbekistan creates ecological and economic problems, increasing genetic erosion of the species [7]. In fact, licorice natural habitats have decreased from 38 thousand hectares in the 1960s to 6.5 thousand hectares in 2020 in Karakalpakstan [8]. This figure shows this plant is in danger of extinction due to excessive harvesting.

Further development of the licorice industry should aim to make more sustainable licorice cultivation that promotes the positive ecological effects on a sustainable approach. In the light of ecological disaster e.g., soil salinity, sandstorms, drought, air pollution in the Aral Sea regions, a primary role of licorice cultivation and restoration of natural habitat should focus in mitigating harsh side effects of ecological severities [9]. The aforementioned facts once more confirmed our assumptions that licorice cultivation in saline lands of the Aral Sea region has a great potential to rehabilitate abandoned saline lands, thereby enhancing the economic and environmental sustainability of the region. In addition, it will introduce climate-resilient agriculture, advance smart farming technologies that increase the productivity of salinity-affected lands and build resilience to climate change.

## 2. Ameliorative Functions of Licorice

Long- and short-term impacts of licorice cultivation in Karakalpakstan have shown that it plays a major role in soil health due to many advantages e.g., protection against erosion, reducing upward salt movement, lowering groundwater levels, improve soil fertility by nitrogen fixation ability and so on. Furthermore, the aboveground part of the licorice plant can reach from up to 0.7–2 m, serving as valuable livestock fodder. Particularly, livestock and poultry play an important role in the livelihood of rural households in Karakalpakstan, which in the past was home to a million heads of sheep, goats and poultry. However, numbers have declined over the years due to insufficient feed and water supply. In addition, natural selection in terms of adaptability to climatic hardship has led to major changes in the composition of Karakalpakstan’s pasture plants, resulting in increased vegetation unsuitable for grazing, lower tender biomass yields, erosion and depletion of pasture quality. Loss of vegetation has also been caused by overgrazing of cattle, cutting of firewood trees and shrubs, discharge of irrigation water into desert depressions and improper irrigation. Extensive experiments were conducted to reveal a role of licorice in enriching nutrition of poultry and livestock when it is used a feed additive [10,11]. Implementing this practice into production is still a long way because it will further facilitate continuously increasing licorice consumption. However, this goal might be achieved by establishing artificial licorice cultivation whenever possible, mostly in increasing degraded lands [12].

Until more recently, research on licorice cultivation focused on soil fertility maintenance, erosion control and water conservation. Conventional approaches of agricultural practices were considered an important tool for improving crop potential and production. Most soils used for long-term licorice planting were improved. Experiments on soil fertility maintenance demonstrated that three years of licorice cultivation enhanced the following wheat and cotton yield considerably in Mirzachul area where soil salinity is at a high level [7]. The cost-benefit analysis also showed that licorice cultivation can be more profitable than cotton and wheat production. In addition, the economic and environmental sustainability of abandoned lands were improved by the land reclamation and remediation functions of this practice [13].

Gafurova and Juliev [14] studied various agro-ecotechnologies to combat crop production challenges in the Aral Sea regions and found that long-term licorice cultivation enriches the soil with organic matter and increases water-resistant aggregates by 70–80%, reduces bulk density up to 1.3–1.4 g/cm^3^ and its roots penetrate to a depth of 3.5–4 m, lowers the saline groundwater. Moreover, symbiotic root-nodulating bacteria fix nitrogen from the air and enrich the soil by increasing total N, P and exchangeable K contents.

Since the 2000s, depending on the interests and capacity of farmers, licorice cultivation in saline lands with other crops create an opportunity full use of these abandoned lands, thereby increasing the economic benefit per unit of land. Under licorice cultivation total dissolved salts declined from 6.35 to 3.99 g L^−1^ over the three years period, indicating the ameliorative bioremediation functions of establishing licorice on the total salt balance. This bioremediation practice of abandoned saline lands may in part be attributed to clean of salts within the root growth profile [15]. A soil erosion control experiment demonstrated reduced soil erosion when licorice was grown on sandy saline land [16]. After going thru a series of analysis on soil salinity, groundwater table and salinity, soil fertility, soil physical properties, these authors of the case study confirmed the improvements in soil conditions together with the profitability of licorice cultivation under severe environments and instable climate surroundings. Research studies by Dagar et al. [17] present phytoremediation and soil properties improvement functions of licorice cultivated in sodic soils where most conventional crops are vulnerable. Having conducted comprehensive soil analysis, these researchers reported that large-scale licorice cultivation could be a solution in rehabilitating degraded soil because its root reaches groundwater level and acts as a bio-drainage system, while the rootstocks enrich soil through nitrogen-fixing bacteria living in them.

Since licorice cultivation offers a potential solution to rehabilitating abandoned saline lands, decision makers should consider providing special incentives e.g., granting special leases for these lands, tax and quota exemption for a given period, provision of soft loans and other such incentives that would facilitate in the recovery of these lands. The priorities of licorice research based on land amelioration have shown promising results regarding phytoremediation and soil salinity reduction; however, further and most detailed studies are needed to implement optimal propagation and cultivation technologies based on innovative techniques.

## 3. Research and Development Initiatives to Enhance Licorice Production

Licorice is considered as a wild plant with good adaptable properties in a wide range of environments, e.g., mean air temperature ranging from 5–25 °C, an annual rainfall between 400 and 1160 mm, and soil pH of 5.7–8.2 [18]. This plant is known to grow well in deep fertile sandy soils near streams in the subtropics [19]. Over the years, studies demonstrated that licorice also grows well in fertile soils. There were laboratory experiments dedicated to study an effect of biochar and soft rock application as an amendment in enhancing licorice yield and quality. It turned out that the improved soil nutrient status could substantially enhance yield quality parameters of licorice [20]. This plant prefers deep and well drained soils. Sandy soils are considered convenient for normal growth of licorice, however, clay soils hinder plant establishment and development [21]. The phytoremediation and soil improvement properties of licorice are considered for cultivation in sodic soils, since most of the conventional crops are sensitive under such conditions [6].

The deep root system allows for more efficient use of water; therefore, licorice plants could be considered as tolerant in water stress conditions and dry climates with relatively lower demand for irrigation water [22]. Apart from infertile soil, adverse environmental conditions e.g., soil salinity, drought, and excessive heat affect crop productivity and quality. Certain licorice genotypes were found to have higher tolerance under these circumstances. For example, *G. glabra* can produce the highest yield of roots under wide range of soil salinity, and even thrives well in dry seasons of warm regions [6]. According to these authors, irrigation of *G. glabra* is a necessary activity in sandy soils of oasis regions to achieve stable yield. Application of licorice cultivation in these lands facilitates in sand stabilization and soil improvement, though inter-plantation of legumes, cereals and vegetables along with licorice is feasible for the first two years under irrigated open field agriculture. The above-ground parts of licorice developed well in 3–4 years after planting and roots are harvested at this period. However, this practice is discouraged in the view of the increase in weed population. Furthermore, some researchers disagreed with intercropping or mixed-cropping practices, raising a concern that licorice plants may become suppressed and root growth severely affected with a consequent effect on total yield within the subsequent years [23,24]. Plant density plays an essential role in total root yield; low density is preferred for active rhizome formation, whereas high density is beneficial for higher root production [25]. Plant density should be around 20,000–24,000 plants ha^−1^, while keeping a space between rows 30–80 cm and 20–50 cm distance within rows [23].

Significant efforts were made to enhance of *G. glabra* and *G. uralensis* seedling multiplication rate by in vitro using micropropagation and colonel propagation methods [26]. These plant tissue culture techniques were substantially effective to enhance seedling multiplication rate of licorice by 15–20 fold when compared to conventional propagation through stolon cuttings [27].

Licorice is usually propagated from cuttings using the potato planting equipment and planting takes place early spring, whereas the optimal size of cuttings should be 15–20 cm in length and 1.5–2.0 cm in diameter [28]. This cultivar also propagates by a sexual way, forming a seed in the generative organs. It can be transplanted when young seedlings are ready to grow in open-field conditions. When compared to sexual propagation with vegetative propagation, a lower plant development is observed at the early growth stage [29].

Since seed germination percentage is low in *Glycyrrhiza* species, seed treatment application must be considered prior to seeding. Licorice seeds covered by hard coat is the reason of the physical dormancy [30]. Therefore, various seed treatment techniques e.g., scarification treatments and temperatures have been studied to increase germination rate and render sexual propagation applicable for commercial licorice cultivation. According to Ghadiri and Torshiz (2000), the germination percentage at 25–35 °C significantly increased when the chemical scarification was used by applying sulfuric acid (70%) for 45 min [31]. More pronounced increase of germination percentage by up to 95% was achieved with sulfuric acid (98%) for 1 h and incubation at 25 and 10 °C and 12 h/12 h light/dark, respectively, whereas mechanical scarification with a hand rotary scarifier improves germination rates at a broader temperature range (15–35 °C). Seed germination is also affected by salinity, with differences in response being observed among the various species of liquorice, and salinity-inducing ions. Moreover, seed germination and seedling growth of *G. uralensis* is significantly affected by pH, where the best results were achieved for values of pH = 6 [32].

Despite the fact that water is needed for normal plant functioning, the water demand of licorice is lower compared to many other crop species. However, previous studies showed that the growth and development of root and rhizomes of the cultivar essentially depends on irrigation [33]. Licorice demands low amount of nutrients as well. For example, growers use 40 kg ha-1 N and 40 kg ha^−1^ P for one licorice growth season [34]. The full portion of N along with the half portion of P are applied as base dressing. The remaining amount of N is used with first or second irrigation.

Having studied licorice production from seeds under saline soils of the Aral See region, Tajetdinov (2021) reported advantages of various biostimulators in combination with mineral fertilizers on the seed germination, biomass accumulation, photosynthetic efficacy, root yield [8]. The highest root yield (12.6 ton ha^−1^) was obtained at the N_100_P_140_K_80_ fertilization which was considerable higher than other mineral fertilization rates. However, in highly saline soils these effects were less perceptible. The author also recommended to sow 5 kg ha^−1^ seeds after soaking for 36 h.

Licorice also benefits from symbiosis with mycorrhiza that substantially improves nutrient uptake, growth and productivity [34]. These authors also reported that the inoculation of licorice plants with arbuscular mycorrhizal fungi enhanced vegetative growth characteristics e.g., leaf area, shoots and root weight as compared to the non-inoculated control, whereas the formulation of secondary metabolites also boosted. Similar results were observed by Chen et al. (2017) exhibiting positive symbiotic effects of arbuscular mycorrhiza on photosynthetic efficiency and flavonoids accumulation [35]. In addition, Orujei et al. (2013) found 9-fold and 3-fold increase in glycyrrhizin content, and a nine-fold and six-fold increase in total phenolics content, respectively) in *G. glabra* roots after the AM inoculation [36]. In recent studies, a group of authors reported the beneficial effects of biochar amendments to improve licorice growth and nutrient uptake under salt stress in the greenhouse [37]. Even though, there were some improvements in terms of acquisition of C (carbon), nitrogen (N), and phosphorus (P) and soil enzyme activities [37], these kinds of experiments far from feasibility under open field agriculture in a large scale.

The harvesting of licorice occurs when the roots reach the marketable size during the autumn after 3–4 years of plant establishment. A total yield of 11,400 kg ha^−1^ can be achieved at a plant density of 25,800 with similar quantities of roots (5460 kg ha^−1^) and rhizomes (5940 kg ha^−1^) [27], whereas, the total yield can be increased up to 15,420 kg ha^−1^ with increase in plant density. Moreover, Marzi et al. informed that at a plant density of 42,000 plants ha^−1^, the root yield might reach 20.4 t ha^−1^, whereas the yield quality, e.g., the total glycyrrhizin content was not significantly affected [28].

## 4. Salt Tolerance Mechanism

Salinity along with drought, extremal temperatures, and low soil fertility are considered the most important abiotic stresses, limiting agricultural productivity. Nonetheless, these mechanisms are still under-researched and should be studied further as a strategic tool to counteract abiotic stresses. Salt tolerance is an important trait employed by the plant to withstand adverse environmental conditions, by modifying short- and long-term adaptation strategies [38]. Genetic improvement is considered as an advanced technique to increase crop stress tolerance, but it requires a long period for a breeding program which follows special cultivation environments and crop performance validation [39]. Hence, scientists have paid more attention to several functional and regulatory genes involved in plant salt tolerance [40]. In addition to these strategies, biostimulants containing bioactive molecules have been proposed to improve plant capability to face adverse environmental conditions. *G. glabra* and *G. uralensis* in association with *Mesorhizobium* strains could fix N2 from the atmosphere after the establishment of effective nodules in their roots [41]. This symbiotic association is vital for host health, promote immune system development, provide nutrients, and protect from pathogens, thereby enhancing plant tolerance to abiotic and biotic stresses [42].

In vitro selection method is also widely used for determining stress tolerance in plants. This culture-based tool effectively determines physiological and biochemical changes in plants under adverse conditions, thereby allowing a better understanding of plant tolerance to salt and drought. The initial process in this method starts with the induction of a genetic variation among cells, tissues or organs. Exposing stressors after some periods reveal the survived cells which will be the subsequent regeneration of the whole organism [43]. These researchers concluded that in vitro selection is a time and resource saving approach compared with classic molecular engineering, however some limitations still exist concerning the stability of the selected traits and epigenetic adaptation.

Many stress-tolerance functions are unknown, and their characterization was still not fully revealed; therefore, they should be studied further in term of their role in plants. Interestingly, some researchers speculated that the reduction of soil salinity might be associated with the absorption of salts by the licorice roots in synthesizing glycyrrhizin acid [16].

## 5. Breeding and Genomics

The genus Glycyrrhiza includes nearly 30 species. Glycyrrhiza species such as Chinese licorice (*Glycorrhiza uralensis* Fisch), Russian licorice (*Glycorrhiza echinate* L.) and *Glycorrhiza inflata* Bat. are widely used in pharma industry and traditional medicine [23]. The quality of raw licorice depends on the level of secondary metabolites, e.g., glycyrrhizin and liquiritin which are considerably differentiated between various glycyrrhiza species. Most notably, these two secondary compounds are in high amount in wild species than cultivated plants of licorice [44]. Therefore, the selection of wild strains with high level of glycyrrhizin is in high importance. Remarkably, the performance of stress-resistant mechanisms in licorice is genotype-dependent. Superior licorice genotypes produce good amount of biomass with high quality (rich in bioactive compounds such as glycyrrhizin), which means high stress tolerance that ensures high yield under harsh conditions. However, most of the previous studies were based on a few genotypes. Advanced genomic and genetic tools have contributed to describing the crop’s genetic diversity. However, in the case of licorice, comprehensive studies should be conducted to reveal a genetic potential and increase the plant’s productivity.

Multi-omics technology has added wings to licorice studies. Recently, the genome of *G. uralensis* and biosynthetic pathway of glycyrrhizin were elucidated [45,46]. Chloroplast genomes of *G. glabra* [47] and *G. lepidota* [48] have been presented. Barcodes for identification of licorice germplasm resources [49] and SSR markers in different interspecific-cross populations [50] have been developed. However, molecular breeding of superior licorice varieties still needs more wild licorice resources as well as molecular markers for elite traits. Extensive collection and evaluation of wild licorice resources will provide basis for licorice breeding. Further studies of molecular mechanisms regulating licorice production and qualities will speed up molecular breeding process.

## 6. Conclusions

Agronomic research is a valuable tool to recover the agricultural land-use systems and biodiversity in the Aral Sea regions, which were seriously damaged under the adverse impacts of anthropogenic factors and climate change. As shown in previous studies, exploration of licorice cultivation technologies and thereby enhancing its cultivated area would be the potential approach in these circumstances. More importantly, implementing advanced technologies and knowledge will empower crop producers to enhance licorice production, thereby leading to sustainable advancement. Sustainability of licorice production is being supported by regional and international efforts that are focused to implementing the best practices and strategies at all levels. Special interest is also highlighted in those practices, e.g., breeding and genomic approaches, which could improve quantity and quality in commercial production. Since land plots with wild licorice are gradually dwindling, the proposed licorice cultivation technologies are likely to have considerable long-lasting benefits for the regeneration of degraded land and ecosystem restoration.

## Figures and Tables

**Table 1 ijerph-19-11770-t001:** Soil salination types in irrigated lands of Karakalpakstan.

Soil Salination Types	Percentage/Hectare	Description
Very high saline soils	15%/69,066 hectare	EC 8–15; pH 8–9; Very high level of Na^+^, K, Ca^2+^, Mg^2+^ and Cl^−^ ions. Poor physical structure, No living habitats.
High saline soils	21%/96,692 hectare	EC 5–8; pH 8–8.5; High level of soluble salts and ions. Poor physical structure, some salt tolerant living habitats, including licorice.
Moderate saline soils	30%/138,132 hectare	EC 2–4; pH 8–8.5; High level of soluble salts and ions. Poor physical structure, some salt tolerant living habitats, including.
Low saline soils	26%/119,714 hectare	EC 1–2; pH 7.5–8.0; Levels of soluble salts and ions exceeds the normal level. Poor physical structure, salt tolerant crops (licorice) and other living habitats.
Normal soils	8%/36,835 hectare	EC 0.75; pH 7–7.5; normal soils for growing any crops.

## Data Availability

Not applicable.

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
