# Peer review of "Perspectives of Licorice Production in Harsh Environments of the Aral Sea Regions"

_ijerph, 2022, doi:10.3390/ijerph191811770_

Round 1
Reviewer 1 Report
The manuscript gives us interesting points on licorice cultivation in the Aral Sea regions. The main topic of the manuscript is the salinization problem of the Aral sea region and problems and solutions to licorice cultivation in saline areas, but no given information about the interaction between licorice and salinity. Although title number 4 has some information about general information on plant salt stress and specific information on licorice x salinity. The same problem has been seen the title number 5, there is no information about licorice and breeding/genomics. Overall the evaluation of the manuscript in my view, the manuscript has to be improved/re-write and can not be published with this version.
Author Response
Thanks. We revised this MS according to your comments. Particularly, necessary contexts were added in the 4 and 5 sections. In addition, the picture showing a catastrophe of Aral Sea drying and the table of soil types were added accordingly.
Thank you for your valuable comments. They really helped to improve the quality of this paper.
Reviewer 2 Report
ijerph-1874446-peer-review-v1
Perspective: Perspectives of licorice production in harsh environments of the Aral Sea regions.
The idea of the article is good and interesting, discusses an important topic related to the importance of licorice plants under saline soil conditions. Despite interesting idea, the manuscript has a lot of flaws that needs to be corrected before consideration for publication. Some suggestions and comments indicated below for author’s further improvement:
Title: Please amend the title of the manuscript with reference to salty environments
Abstract: The abstract is well written. But line 22 please remove "drought".
Introduction:
Line 47-49: The sentence is too complex and unnecessary or simplified please
Paragraph (line 58-62) needs to add references.
Line 50: The Aral Sea disaster ???
Also the paragraph (line 50-57) needs to be reformulated, and appropriate references added.
The paragraph from line 79-80 is incomprehensible.
Please rephrase the purpose of the article.
Paragraph 119-124: Gafurova and Juliev (2021) studied various agro-ecotechnologies to combat crop production challenges in the Aral Sea regions and found that long-term licorice cultivation enriches the soil with organic matter and increases water-resistant aggregates by 70–80%, 121 reduces bulk density up to 1.3–1.4 g/cm3 and its roots penetrate to a depth of 3.5–4 m, 122 lowers the saline groundwater [14]. Reformulated to "Gafurova and Juliev [14] studied various agro-ecotechnologies to combat crop production challenges in the Aral Sea regions and found that long-term licorice cultivation enriches the soil with organic matter and increases water-resistant aggregates by 70–80%, 121 reduces bulk density up to 1.3–1.4 g/cm3 and its roots penetrate to a depth of 3.5–4 m, 122 lowers the saline groundwater".
Paragraph 137-140: Research studies by Dagar et al. (2015) present phytoremediation and soil properties improvement functions of licorice cultivated in sodic soils where most conventional crops are vulnerable [17]. Reformulated to "Research studies by Dagar et al. [17] present phytoremediation and soil properties improvement functions of licorice cultivated in sodic soils where most conventional crops are vulnerable. "
Paragraph 144-147: unnecessary.
Line 150: The authors mentioned innovative techniques, what is it and is it not found in recent research.
Line 163-165: Add appropriate reference.
Paragraph (Line 183-187) inconsistent with paragraph (Line 192-198) should be merged into one paragraph, As well as with the next paragraph to them (Line 199-213).
In the article, the authors focused on paragraphs related to methods of cultivation, fertilization, sexual propagation, vegetative propagation, the physical dormancy, fungi, and others. It was more likely to focus on one factor, which is salinity, for example.
Line 245: remove "according to Douglas et al. (2004)"
Paragraph (line 239-241) needs to add references.
The paragraph (line 267-276) is unclear and needs to be reformulated and further clarified.
Conclusions is well written.
Author Response
Response to reviewer’s comments
________________________________________________________________
We would like thank to the reviewer for valuable comments and suggestions as well as targeted criticisms and good advice. All these served to improve the contents of the paper. Response to each reviewer’s comments were listed below:
Perspective: Perspectives of licorice production in harsh environments of the Aral Sea regions.
The idea of the article is good and interesting, discusses an important topic related to the importance of licorice plants under saline soil conditions. Despite interesting idea, the manuscript has a lot of flaws that needs to be corrected before consideration for publication. Some suggestions and comments indicated below for author’s further improvement:
Title: Please amend the title of the manuscript with reference to salty environments
Abstract: The abstract is well written. But line 22 please remove "drought".
Thank you for these comments. The indicated mistake was revised as per your suggestion.
Introduction:
Line 47-49: The sentence is too complex and unnecessary or simplified please
Thank you for these comments. The mistakes were revised as per your suggestions.
Paragraph (line 58-62) needs to add references.
Thank you for these comments. The mistakes were revised as per your suggestions.
Line 50: The Aral Sea disaster ???
Also the paragraph (line 50-57) needs to be reformulated, and appropriate references added.
Thank you for your comments. The indicated mistakes were revised and appropriate reference was added.
The paragraph from line 79-80 is incomprehensible.
Thank you for these comments. The mistakes were revised as per your suggestions.
Please rephrase the purpose of the article.
Paragraph 119-124: Gafurova and Juliev (2021) studied various agro-ecotechnologies to combat crop production challenges in the Aral Sea regions and found that long-term licorice cultivation enriches the soil with organic matter and increases water-resistant aggregates by 70–80%, 121 reduces bulk density up to 1.3–1.4 g/cm3 and its roots penetrate to a depth of 3.5–4 m, 122 lowers the saline groundwater [14]. Reformulated to "Gafurova and Juliev [14] studied various agro-ecotechnologies to combat crop production challenges in the Aral Sea regions and found that long-term licorice cultivation enriches the soil with organic matter and increases water-resistant aggregates by 70–80%, 121 reduces bulk density up to 1.3–1.4 g/cm3 and its roots penetrate to a depth of 3.5–4 m, 122 lowers the saline groundwater".
Thank you for these comments. The mistakes were revised as per your suggestions.
Paragraph 137-140: Research studies by Dagar et al. (2015) present phytoremediation and soil properties improvement functions of licorice cultivated in sodic soils where most conventional crops are vulnerable [17]. Reformulated to "Research studies by Dagar et al. [17] present phytoremediation and soil properties improvement functions of licorice cultivated in sodic soils where most conventional crops are vulnerable. "
Thank you for these comments. The mistakes were revised as per your suggestions.
Paragraph 144-147: unnecessary.
Thank you for these comments. We decided to leave this context, so it gives some suggestions to enhance licorice cultivation in the region.
Line 150: The authors mentioned innovative techniques, what is it and is it not found in recent research.
Thanks. Innovative techniques such as in-vitro propagations using micropropagation and colonel propagation methods, seed inoculations with efficient bacteria, breeding techniques and many others were discussed in the next paragraphs.
Line 163-165: Add appropriate reference.
Thank you for these comments. The mistakes were revised as per your suggestions.
Paragraph (Line 183-187) inconsistent with paragraph (Line 192-198) should be merged into one paragraph, As well as with the next paragraph to them (Line 199-213).
In the article, the authors focused on paragraphs related to methods of cultivation, fertilization, sexual propagation, vegetative propagation, the physical dormancy, fungi, and others. It was more likely to focus on one factor, which is salinity, for example.
Thank you for these comments. All these agrotechnologies are important solutions to combat salt stress in these regions. Unfortunately, these solutions were not used so far and our goal is to show for crop producers and implement them in licorice production.
Line 245: remove "according to Douglas et al. (2004)"
Thank you. The mistakes were corrected as per your suggestions.
Paragraph (line 239-241) needs to add references.
Thanks. The necessary reference was added.
The paragraph (line 267-276) is unclear and needs to be reformulated and further clarified.
Conclusions is well written.
Thank you for your valuable comments. They really helped to improve the quality of this paper.

Round 2
Reviewer 1 Report
Authors improved the my concern about the manuscript in the new version. They added more information about breeding and genomics perspectives.
Reviewer 2 Report
The manuscript has been significantly improved
It can be accepted in the present form
All the best wishes